# BridgeRAG: A Framework for Reasoning over Partitioned Knowledge Graphs

## Abstract

Existing Knowledge Graph-based RAG (Retrieval-Augmented Generation) systems face a fundamental dilemma in multi-document scenarios. They either treat each document as an isolated knowledge graph, which preserves contextual purity but prevents cross-document reasoning, or merge them into a single, massive graph, leading to entity saturation and contextual noise pollution. To resolve this core conflict, we introduce the BridgeRAG framework, designed to elegantly achieve both "partitioned isolation" and "cross-partition linking" for multiple documents. BridgeRAG is a collaborative framework that integrates static linking and dynamic reasoning. Experiments on multi-hop question answering benchmarks like HotpotQA show that BridgeRAG significantly outperforms state-of-the-art RAG models, especially on complex questions that require deep cross-partition navigation.

## 1 INTRODUCTION

Despite the remarkable capabilities of Large Language Models (LLMs) Brannon et al. (2023); Chen et al. (2024), their reliance on static knowledge and susceptibility to factual hallucinations limit their application. To address this, Retrieval-Augmented Generation (RAG) has emerged as a paradigm that grounds LLMs in external knowledge bases Lewis et al. (2020), supplementing them with up-to-date or private data. Knowledge Graph-based RAG (KG-RAG) further enables more precise multi-step reasoning by leveraging structured information Yu et al. (2024); Gao et al. (2025).

However, applying KG-RAG to real-world, multi-document scenarios exposes a fundamental dilemma: either constructing an independent knowledge graph for each document Mao et al. (2025), which preserves contextual purity but has a fatal flaw in its inability to reason across documents when faced with questions requiring the integration of multi-source information; or merging all documents into a single, monolithic graph Baek et al. (2023); Kang et al. (2023). This latter approach leads to "entity saturation" and contextual noise pollution, especially when entity alignment is ambiguous (e.g., "Apple" the company versus "apple" the fruit). This contamination of the semantic structure ultimately causes the system to retrieve a large volume of irrelevant or even contradictory information, thereby misleading the LLM's judgment.

The core solution to this conflict lies in skillfully achieving both "partitioned isolation" and "cross-partition linking." Chan et al. (2024); Fan et al. (2025) This paper introduces BridgeRAG, a novel iterative reasoning framework designed to build bridges across these isolated knowledge partitions. Our approach actualizes an effective and efficient RAG system by addressing three key challenges: (C1) How to intelligently connect independent document knowledge graphs? (C2) How to precisely retrieve relevant documents for a cross-document query? (C3) How to focus on highly relevant information within the retrieved documents while avoiding noise?

BridgeRAG systematically addresses these challenges through a multi-component, synergistic architecture.Our approach is predicated on a key observation: the semantic core of a document is anchored by its named entities, such as specific people, places, and events. Consequently, any meaningful relationship between two documents is established through these shared entities. Even when a link is indirect, it is invariably mediated by an intermediary document that shares distinct named entities with each of the original documents. Therefore, we posit that shared named entities form the fundamental conduits for establishing and navigating inter-document relationships.

Based on this premise, we have designed: 1) A dual-verification mechanism combining an LLM and an embedding model to construct high-fidelity SAME_AS cross-document entity links, addressing (C1). 2) A multi-source weighted routing module that focuses on a small set of key documents by evaluating the relevance of their summaries and entity lists, addressing (C2). 3) An iterative reasoning engine driven by an LLM agent, which dynamically constructs a reasoning path by generating sub-questions and leverages a Dynamic Working Memory (DWM) mechanism to manage context, addressing (C3).

To address the core dilemma of contextual purity versus knowledge connectivity in multi-document KG-RAG, we propose **BridgeRAG**, a novel framework based on a synergistic two-phase architecture. We first perform offline **knowledge pre-digestion** and **hybrid entity linking** to build a robust navigational backbone across isolated knowledge partitions. Then, an online agent performs dynamic, iterative reasoning guided by a **Dynamic Working Memory (DWM)**. The contributions are listed as follows:

- We propose **BridgeRAG**, a novel paradigm for multi-document RAG that resolves the conflict between contextual purity and knowledge connectivity.
- We design a dynamic reasoning agent empowered by three key mechanisms: **knowledge pre-digestion**, **hybrid entity linking**, and a **Dynamic Working Memory (DWM)**.
- Extensive experiments and analyses on multi-hop QA benchmarks verify the superiority of our proposed **BridgeRAG** framework.

## 2 RELATED WORK

### 2.1 RAG TO KNOWLEDGE GRAPH RAG

Retrieval-Augmented Generation (RAG) has become a central paradigm for mitigating the factual hallucinations and knowledge obsolescence issues inherent in Large Language Models (LLMs). This paradigm has evolved from early sparse retrievers (e.g., BM25) to dense retrievers (e.g., DPR) capable of superior semantic matching. DPR learns to embed questions and passages into a shared vector space, thereby enabling more effective semantic alignment Zhou et al. (2025).

The insufficiency of one-shot retrieval for multi-hop queries led to iterative methods like **IRCoT** Trivedi et al. (2022). While they expand context, their navigation of unstructured text is imprecise and lacks an understanding of the knowledge structure. Autonomous systems like **Self-RAG** Asai et al. (2024) introduced self-awareness with "reflection tokens", but their effectiveness remains capped by the challenge of reasoning over unstructured text. This core limitation of all text-based RAG methods makes adoption of Knowledge Graphs (KGs) a necessary evolution. KGs provide the explicit structure and relational pathways required for precise, logical reasoning.

### 2.2 CHALLENGES IN MULTI-DOCUMENT KGS

Despite its promise, applying KG–RAG to real-world, multi-document corpora exposes a fundamental architectural dilemma, as illustrated in Figure 1. Existing strategies are trapped in a trade-off. One approach is to construct isolated **Partitioned Knowledge Graphs (PKGs)** Han et al. (2025); Luo et al. (2025), which preserves contextual purity but creates information silos that make cross-document reasoning impossible. The opposing strategy creates a single **Monolithic Knowledge Graph (MKG)** Hu et al. (2019); Matsumoto et al. (2024), a static "aggregate-first, query-later" model that ensures connectivity but inevitably suffers from fatal flaws like "entity saturation" and contextual noise from ambiguous linking Zhang et al. (2022). Our work, **BridgeRAG**, is designed to break this stalemate by synergizing the advantages of both isolation and linking.

### 2.3 LIMITATIONS OF LLM'S GRAPH NAVIGATION

Recent advancements have positioned LLMs as reasoning agents Singh et al. (2025); Wei et al. (2022); Yao et al. (2023), but their inherently text-based nature makes them inefficient at navigating knowledge graphs, often resulting in brute-force search rather than precise traversal. While agentic systems like Self-RAG Asai et al. (2024); Xiong et al. (2025) add self-reflection, their focus is on

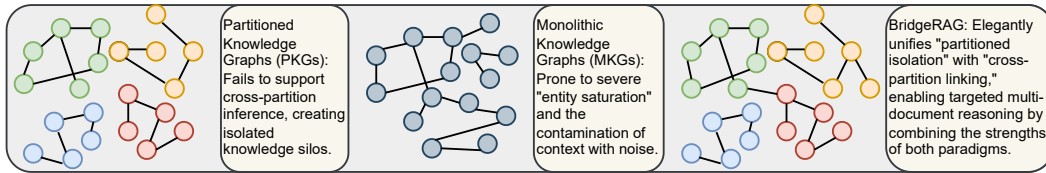

Figure 1: **The architectural dilemma in multi-document KG-RAG. (Left)** Partitioned KGs (PKGs) ensure contextual purity but create isolated knowledge silos. **(Center)** Monolithic KGs (MKGs) enable connectivity but suffer from entity saturation and noise. **(Right) BridgeRAG** unifies both, using a navigational backbone to enable precise reasoning across partitions.

evaluating isolated information. In contrast, BridgeRAG proactively constructs a reasoning path within a structured knowledge space. It moves beyond mere evaluation by decomposing a problem into targeted sub-questions to navigate the multi-graph environment, exhibiting a deeper capacity for the structured planning crucial for multi-hop reasoning.

## 3 METHODOLOGY

### 3.1 OVERALL FRAMEWORK AND DESIGN

The core of our proposed BridgeRAG framework is a synergistic two-phase architecture, designed to fundamentally address the challenges LLM Agents face when reasoning over complex, multi-partitioned knowledge bases. Our design philosophy is rooted in a recognition of the intrinsic challenges LLMs face with graph-structured data. The worldview of general-purpose agents is inherently text-based; they lack a native understanding of graph topology. Therefore, instead of forcing the agent to adapt to the data, we reshape the knowledge to adapt to the agent.

To this end, we designed a two-stage summarization process to "pre-digest" structured graph information into the natural language format that LLMs excel at processing. First, for each core named entity $e$ within a document's knowledge graph, we generate an information-dense **Entity Summary**, $S_e$. This is achieved by prompting an LLM, $f_{sum}$, to synthesize the entity's description, $\text{Desc}(e)$, with all of its first-degree relational information, $N_1(e)$:

$$S_e = f_{sum}(\text{Desc}(e), N_1(e)). \tag{1}$$

This crucial "knowledge pre-digestion" step effectively flattens a local graph structure into a coherent natural language text. Subsequently, we generate a higher-level **Document Summary**, $S_d$, by prompting an aggregation LLM, $f_{agg}$, to summarize the set of all core entity summaries $\{S_{e_i}\}$ within that document:

$$S_d = f_{agg}(\{S_{e_1}, S_{e_2}, \ldots, S_{e_k}\}). \tag{2}$$

This summary encapsulates the document's core information, providing a powerful signal for retrieval during online inference. Based on this principle, the BridgeRAG workflow is divided into two phases:

**Offline Knowledge Base Construction.** We begin by constructing independent Partitioned Knowledge Graphs (Partitioned KGs) from each document to ensure contextual purity (partitioned isolation). Subsequently, we perform knowledge pre-digestion to generate entity and document summaries. Finally, a Hybrid Entity Linking mechanism establishes a high-fidelity cross-graph navigational backbone, laying the groundwork for cross-partition linking.

**Online Iterative Reasoning.** When a query $q$ arrives, the Agent interacts with the pre-digested document summaries $\{S_{d_i}\}$ rather than traversing raw graphs. It first uses a Multi-Source Weighted Router to generate a retrieval plan $P = (D_p, D_a)$, partitioning relevant documents into Primary ($D_p$) and Auxiliary ($D_a$) sets. The agent then enters an iterative reasoning loop managed by a Dynamic Working Memory (DWM), whose state at iteration $t$ is denoted by $M_t$. The process starts with $M_0 = \{q\}$. In each iteration, the agent executes an action to gather new evidence $Ev_t$, updating the memory as:

$$M_{t+1} = M_t \cup Ev_t. \tag{3}$$

This loop of proactive planning and guided retrieval continues until a termination condition is met, at which point the agent synthesizes the final answer $A$ from the complete context $M_{\text{final}}$ using a generation function $f_{\text{synth}}$.

This model, which combines "offline knowledge reshaping" with "online lightweight navigation," allows BridgeRAG to elegantly unify "partitioned isolation" and "cross-partition linking," achieving efficient and precise multi-document reasoning.

## 3.2 KNOWLEDGE BASE CONSTRUCTION

To support efficient online inference, we designed an offline pipeline that transforms raw documents into a structured and navigable federated knowledge base, as illustrated in Figure 2.

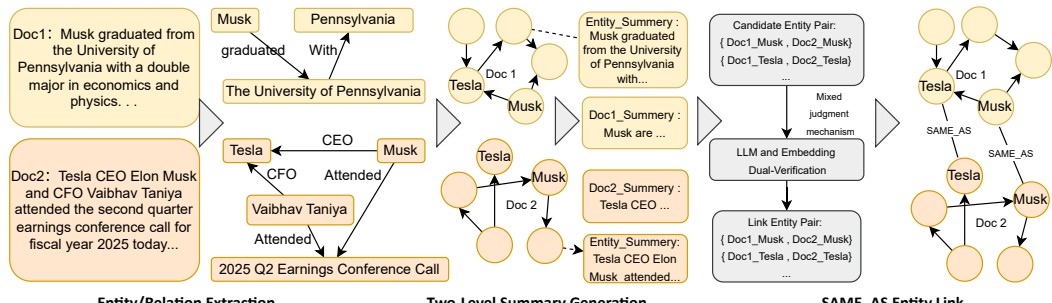

Figure 2: **The Offline Knowledge Base Construction Phase of BridgeRAG.** The process unfolds in three main stages. First, we perform **Entity/Relation Extraction** to build an independent, contextually pure knowledge graph for each document. Second, we execute a **Two-Level Summary Generation** step, creating information-dense summaries for both individual entities and the entire document. Finally, based on these rich summaries, we establish high-fidelity **SAME_AS Entity Links** across different partitions by employing a dual-verification mechanism that leverages both LLMs and embedding models. This entire offline process transforms raw text into a structured, navigable, and LLM-friendly federated knowledge base.

### 3.2.1 PARTITIONED KG CONSTRUCTION AND PRE-DIGESTION

We first process each document $d_i \in \mathcal{D}$ through an independent NLP pipeline, constructing an initial partitioned knowledge graph $G_i = (\mathcal{E}_i, \mathcal{R}_i)$ via Named Entity Recognition (NER) and Relation Extraction (RE).

To make this structured information more accessible to the LLM Agent, we introduce the crucial step of **Knowledge Pre-Digestion**. For each core named entity $e_k \in \mathcal{E}_i$, we generate a concise entity summary $S_{e_k}$ Lyu et al. (2025). This summary is produced by a powerful LLM, denoted by the function $f_{sum}$, which synthesizes the entity's description, $\text{Desc}(e_k)$, with its set of first-degree relational triples, $N_1(e_k)$:

$$S_{e_k} = f_{sum}(\text{Desc}(e_k), N_1(e_k)). \tag{4}$$

This summary encapsulates the entity's core information and provides rich context for cross-document identity verification, forming the basis for disambiguation.

### 3.2.2 CROSS-PARTITION NAVIGATIONAL BACKBONE

After achieving partitioned isolation, to solve challenge (C1) and enable cross-partition linking, we designed a **Hybrid Entity Linking** mechanism to construct a navigational backbone composed of SAME_AS relations. This mechanism performs a dual-verification process on same-name entity pairs $(e_i, e_j)$, where $e_i \in G_i$ and $e_j \in G_j$.

**Efficient Semantic Matching ($S_{\text{emb}}$).** We first use an advanced sentence embedding model, $\mathcal{M}_{emb}$ : string $\rightarrow \mathbb{R}^d$, to encode the entity summaries $S_{e_i}$ and $S_{e_j}$ into high-dimensional vectors.

The semantic similarity score is then computed as the cosine similarity between these vectors:

$$S_{\text{emb}}(e_i, e_j) = \frac{\mathcal{M}_{emb}(S_{e_i}) \cdot \mathcal{M}_{emb}(S_{e_j})}{\|\mathcal{M}_{emb}(S_{e_i})\| \cdot \|\mathcal{M}_{emb}(S_{e_j})\|}. \tag{5}$$

This step allows for the rapid, large-scale filtering of semantically irrelevant entity pairs.

**Deep Semantic Adjudication ($S_{\textbf{LLM}}$).** Candidates that pass an initial similarity filter are submitted to a more powerful LLM, $f_{adj}$, for deep semantic adjudication. The LLM is prompted to act as a "domain expert," leveraging its extensive world knowledge and the context from both entity summaries to determine if they refer to the same real-world entity. It outputs a calibrated confidence score:

$$S_{\text{LLM}}(e_i, e_j) = f_{adj}(S_{e_i}, S_{e_j}) \quad \in [0, 1]. \tag{6}$$

The final link score, $S_{\text{link}}$, is a weighted combination of these two scores, formulated as:

$$S_{\text{link}}(e_i, e_j) = \alpha \cdot S_{\text{LLM}}(e_i, e_j) + (1 - \alpha) \cdot S_{\text{emb}}(e_i, e_j). \tag{7}$$

where $\alpha \in [0, 1]$ is a balancing hyperparameter. A weighted SAME_AS edge with weight $w = S_{\text{link}}(e_i, e_j)$ is established between the two entities if and only if $S_{\text{link}}(e_i, e_j) > \tau$, where $\tau$ is a predefined linking threshold. This process ensures the high fidelity of our navigational backbone.

### 3.2.3 DOCUMENT-LEVEL REPRESENTATION GENERATION

To efficiently address challenge (C2): locating relevant documents precisely during online inference, we create an information-rich representation for each document $d_i$, termed the **Document Manifest**, $\mathcal{M}_i$. It consists of two components: a Named Entity Manifest, $\mathcal{E}_i^{\text{core}} \subseteq \mathcal{E}_i$, which is the set of all core named entities in the document, and a Document Summary, $S_{d_i}$. The latter is a higher-level summary generated by an aggregation LLM, $f_{agg}$, which synthesizes the set of all entity summaries $\{S_{e_k} | e_k \in \mathcal{E}_i^{\text{core}}\}$:

$$S_{d_i} = f_{agg}(\{S_{e_k} | e_k \in \mathcal{E}_i^{\text{core}}\}). \tag{8}$$

The complete Document Manifest is thus defined as:

$$\mathcal{M}_i = (\mathcal{E}_i^{\text{core}}, S_{d_i}). \tag{9}$$

This manifest provides a powerful and precise input signal for the multi-source routing module in the online phase, enabling rapid relevance ranking.

## 3.3 ONLINE ITERATIVE REASONING

To enable precise and efficient multi-document question answering, we designed an online iterative reasoning workflow, which is driven by an LLM agent to perform lightweight navigation over the pre-constructed federated knowledge base. As illustrated in Figure 3, this workflow is centered around a multi-source weighted routing module and an iterative reasoning engine: the routing module first locates a small set of key documents based on the relevance between the query and document summaries (addressing C2), after which the reasoning engine constructs a coherent reasoning chain and manages context by dynamically generating sub-questions and leveraging the Dynamic Working Memory (DWM) mechanism (addressing C3).

### 3.3.1 PLANNING AND CONTEXT CONSTRUCTION

To address challenge (C2) and avoid noise, the initial step of the reasoning process is to locate a high-relevance candidate set from the entire corpus $D$. We designed a multi-source weighted routing mechanism Wang et al. (2023); He et al. (2025) that combines two signals for ranking: 1) the degree of entity matching between the query $Q$ and each document's "Named Entity Manifest," and 2) the semantic similarity between $Q$ and each "Document Summary." By employing the Reciprocal Rank Fusion (RRF) algorithm, we generate a unified, high-relevance candidate set.

The Agent leverages an LLM to generate a detailed Guided Retrieval Plan by taking the original query $Q$ and the summary information of $D_{\text{cand}}$ as input. This plan categorizes $D_{\text{cand}}$ into two types: Primary Documents ($D_{\text{main}}$), one or two documents deemed most central and directly relevant to answering the query, and auxiliary documents ($D_{\text{aux}}$), which contain key supplementary information, with a clear specification of the exact entities to be investigated.

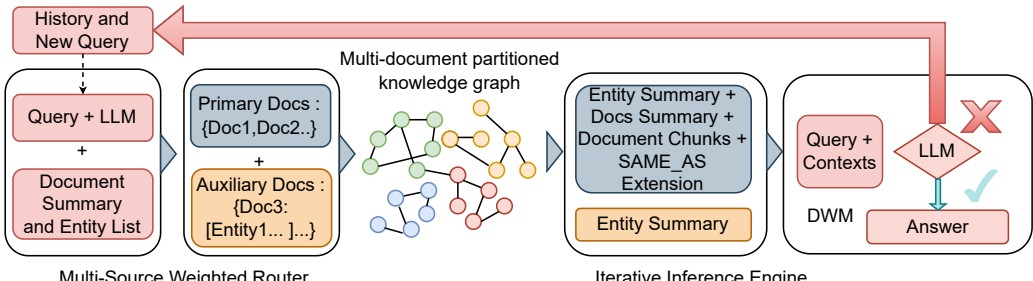

Figure 3: **The Online Iterative Reasoning Phase of BridgeRAG.** The workflow begins when the **Multi-Source Weighted Router**, guided by the query and document summaries, generates a retrieval plan by categorizing relevant documents into **Primary** and **Auxiliary Docs**. The **Iterative Inference Engine** executes a targeted information extraction strategy: it performs in-depth mining on primary docs (extracting summaries, chunks, and performing SAME_AS expansion) and surgical extraction on auxiliary docs (extracting only specified entity summaries). All gathered evidence is loaded into the **Dynamic Working Memory (DWM)**, where the LLM agent enters a self-correction loop, refining its query based on the current context until a sufficient answer can be generated.

### 3.3.2 DYNAMIC WORKING MEMORY (DWM)

Based on the retrieval plan $P_{\text{guide}}$, we extract information selectively. For primary documents $D_{\text{main}}$, we employ an in-depth mining strategy, denoted as $\texttt{Extract}_{\text{main}}$. For auxiliary documents $D_{\text{aux}}$, we use a surgical precision strategy, $\texttt{Extract}_{\text{aux}}$. This dual-pronged approach constructs a high-relevance, low-noise initial context, $C_0$, which is then loaded as the initial state of the Dynamic Working Memory (DWM). This process can be formulated as:

$$C_0 = \texttt{Extract}_{\text{main}}(D_{\text{main}}, Q) \cup \texttt{Extract}_{\text{aux}}(D_{\text{aux}}, P_{\text{guide}}). \tag{10}$$

where $Q$ is the original query.

### 3.3.3 CORE REASONING LOOP AND REFINEMENT

After obtaining the initial context $C_0$, the LLM Agent enters the core loop to tackle challenge (C3). At the beginning of iteration $t$ (for $t \geq 1$), the Agent first evaluates if the context gathered so far, $C_{t-1}$, is sufficient to answer the original query $Q$ using a critique function, IsSufficient$(C_{t-1}, Q)$ Liu et al. (2025). If true, the loop terminates and a final answer is generated.

If the information is insufficient, the Agent executes **Query Refinement**. This process generates a new, more targeted query $Q^{(t)}$ by summarizing the current findings and formulating a precise sub-question to fill the most critical information gap:

$$Q^{(t)} = \texttt{Refine}(Q, C_{t-1}) = Q_{\text{sum}}^{(t)} \oplus Q_{\text{sub}}^{(t)}. \tag{11}$$

where $Q_{\text{sum}}^{(t)}$ is the summary of current findings, $Q_{\text{sub}}^{(t)}$ is the new sub-question, and $\oplus$ denotes string concatenation.

This refined query $Q^{(t)}$ drives the next round of focused planning and extraction, yielding new contextual information, $\Delta C_t$. This new evidence is then appended to the DWM to form the context for the next iteration:

$$C_t = C_{t-1} \cup \Delta C_t. \tag{12}$$

If an answer is not formed within a preset maximum number of iterations, $T_{\text{max}}$, the process terminates. Through this iterative model, which establishes a foundation via "planning-and-extraction" and is driven by a "critique-and-refine" core engine, BridgeRAG achieves robust and dynamic reasoning capabilities.

# 4 EXPERIMENT

## 4.1 EXPERIMENTAL SETUP

We designed a series of experiments to evaluate the effectiveness of BridgeRAG on multi-document, multi-hop question answering tasks, aiming to answer the following research questions: **(RQ1)** How does the end-to-end performance of BridgeRAG compare to state-of-the-art RAG baselines? **(RQ2)** What are the individual contributions of the core components in our framework, such as planning, iteration, summaries, and linking? **(RQ3)** What specific advantages does BridgeRAG demonstrate across various scenarios, as illustrated by case studies? **(RQ4)** How do key hyperparameters, such as those for entity linking and context expansion, affect BridgeRAG's performance?

**Evaluation Datasets:** We evaluate our framework on three challenging datasets Zhang et al. (2025). A meticulously constructed HotpotQA-Subset targets the core multi-hop (2-3 hops) reasoning across documents Yang et al. (2018). The LongBench (QA part) probes the robustness of our summarization and DWM on extremely long contexts (>5000 tokens) Bai et al. (2023). Finally, the 2WikiMultiHopQA-Subset stress-tests the model's long-chain planning and iterative abilities by requiring information integration across a larger number of documents (average of 3-4).

**Evaluation Metrics:** We designed a dual-metric system to assess performance across two key dimensions. For Answer Quality, we evaluate the final generated text using ROUGE-L (for fluency and content overlap), Substring Exact Match (SUB_EM), and token-level F1, Precision, and Recall Şakar & Emekci (2025). For Retrieval Quality, we evaluate the core retrieval module's performance using Retrieval Precision, which measures the amount of noise introduced, and Retrieval Recall, which measures the system's ability to find all relevant evidence.

**Baselines:** We compare BridgeRAG against three representative categories of baselines. Naive RAG, a standard single-pass retrieval model Qian et al. (2024).Iterative Text Retrieval Models, including IRCoT Trivedi et al. (2022)and Iter-RetGen Kumar et al. (2025), which conduct multiple rounds of unstructured text retrieval.Knowledge Graph RAG, represented by GraphRAG Edge et al. (2024), which builds and traverses a monolithic knowledge graph.This comparison is designed to comprehensively position BridgeRAG's advantages within the existing RAG landscape.

For reproducibility, we provide a comprehensive description of all hyperparameter configurations and implementation details in the Appendix.

## 4.2 RESULTS AND ANALYSIS (RQ1)

To answer our first research question (RQ1: How does the end-to-end performance of BridgeRAG compare to state-of-the-art RAG baselines?), we conducted a comprehensive evaluation of BridgeRAG and all baseline models on the HotpotQA-Subset, LongBench and 2WikiMultiHopQA-Subset datasets. The experimental results are presented in Table 1.

The performance of **GraphRAG** provides a compelling validation for our thesis on "entity saturation." While achieving a high ROUGE-L score (56.13%), its monolithic graph introduced massive contextual noise, causing a collapse in TOKEN_PRECISION (39.73%) and a poor F1 score (51.47%). **BridgeRAG** avoids this pitfall, using high-precision retrieval to deliver a focused context and achieve superior F1 performance (65.34%). Furthermore, traditional iterative models (**IRCoT**, **Iter-RetGen**) consistently underperformed, confirming that unstructured iteration fails on complex multi-hop tasks. On the most challenging dataset, **2WikiMultiHopQA**, no baseline surpassed a 55% F1 score. Here, **BridgeRAG**'s guided navigation via SAME_AS links was decisive, achieving a 69.00% F1 score, a commanding lead of over 14 points that proves the necessity of our approach as reasoning complexity increases. Finally, **BridgeRAG** demonstrated strong robustness across diverse challenges. Its leading F1 score (67.47%) on the long-context **LongBench** dataset validates the effectiveness of our summarization and DWM mechanisms. This shows our architecture consistently delivers high-quality answers, whether facing the information overload of long documents or the complex dependencies of multi-hop reasoning.

Table 1: **Overall performance comparison on multi-hop question answering benchmarks.** The **bold** values denote the best results among all models for each specific dataset and metric. BridgeRAG consistently and significantly outperforms most baseline models, demonstrating the superiority of our proposed paradigm.

| Model | Datasets | Answer Quality | | | | | Retrieval Quality | |
|---|---|---|---|---|---|---|---|---|
| | | ROUGE-L | SUB_EM | TOKEN_PRECISION | TOKEN_RECALL | F1 | Retrieval Precision | Retrieval Recall |
| Naive RAG | HotpotQA-Subset | 45.77% | 50.46% | 55.24% | 59.00% | 54.29% | 67.11% | 85.10% |
| | LongBench | 43.28% | 50.61% | 54.37% | 58.76% | 57.78% | 71.47% | 84.90% |
| | 2WikiMultiHopQA-Subset | 43.44% | 51.35% | 48.20% | 51.01% | 47.04% | 62.48% | 78.72% |
| IRCoT | HotpotQA-Subset | 50.83% | 48.42% | 55.71% | 57.22% | 54.23% | 67.09% | 85.08% |
| | LongBench | 45.95% | 53.06% | 55.44% | 59.16% | 56.50% | 73.15% | 88.34% |
| | 2WikiMultiHopQA-Subset | 48.76% | 52.03% | 46.74% | 52.45% | 49.82% | 62.48% | 78.72% |
| Iter-RetGen | HotpotQA-Subset | 44.74% | 50.56% | 54.43% | 59.09% | 53.59% | 69.66% | 88.03% |
| | LongBench | 45.12% | 49.22% | 53.95% | 60.81% | 59.47% | 72.50% | 85.73% |
| | 2WikiMultiHopQA-Subset | 37.22% | 56.08% | 52.51% | 55.86% | 51.50% | 65.63% | 83.45% |
| GraphRAG | HotpotQA-Subset | **56.13%** | 53.28% | 39.73% | 64.52% | 51.47% | 74.87% | 86.55% |
| | LongBench | **60.26%** | 54.87% | 41.19% | 65.77% | 53.31% | 75.49% | 89.64% |
| | 2WikiMultiHopQA-Subset | **53.77%** | 56.81% | 44.16% | 64.49% | 54.71% | 66.18% | 84.09% |
| **BridgeRAG (Ours)** | HotpotQA-Subset | 54.29% | **60.47%** | **66.51%** | **69.42%** | **65.34%** | **79.62%** | **89.15%** |
| | LongBench | 60.07% | **59.92%** | **64.85%** | **69.27%** | **67.47%** | **81.17%** | **90.31%** |
| | 2WikiMultiHopQA-Subset | 46.53% | **72.30%** | **68.51%** | **71.73%** | **69.00%** | **73.11%** | **88.65%** |

## 4.3 ABLATION STUDY (RQ2)

To answer our second research question (RQ2: What are the respective contributions of the core components of the framework, such as planning, iteration, summaries, and linking?), and to quantitatively evaluate the contribution of each core component to the final performance, we conducted a series of rigorous ablation studies. Using the full BridgeRAG model as the baseline, we constructed different variants by independently removing or replacing a key module. As the overall results in Table 2 show, removing any of our designed core components leads to a significant decline in the final answer quality, measured primarily by F1 and SUB_EM scores.

Table 2: **Ablation study results on the HotpotQA-Subset.** The removal of any core component leads to a significant degradation in performance across all evaluated metrics, validating the necessity and synergistic design of our framework.

| Variant | ROUGE-L | SUB_EM | TOKEN_PRECISION | TOKEN_RECALL | F1 | Retrieval Precision | Retrieval Recall |
|---|---|---|---|---|---|---|---|
| **BridgeRAG (Ours)** | **54.29%** | **60.47%** | **66.51%** | **69.42%** | **65.34%** | 79.62% | **89.15%** |
| w/o Iteration | 51.71% | 55.86% | 59.93% | 63.49% | 62.62% | **80.33%** | 86.09% |
| w/o Planning | 48.19% | 52.88% | 56.16% | 62.98% | 60.14% | 71.43% | 84.28% |
| w/o Summaries | 50.84% | 53.19% | 57.21% | 63.82% | 61.01% | 79.36% | 88.75% |
| w/o SAME_AS | 52.39% | 57.75% | 57.11% | 63.12% | 61.97% | 73.98% | 88.47% |

The ablation study confirms the critical role of each component. Removing the **Planning** module caused the most severe performance drop (over 5% in F1), proving that without this top-level guidance, the agent becomes a blind information gatherer, overwhelmed by initial noise. While removing **Iteration** and **SAME_AS linking** had a lesser impact on retrieval, the sharp F1 score decline reveals they are essential for transforming retrieved facts into a coherent answer; they are the bridge from "information gathering" to "deep reasoning." Finally, the performance hit from removing **Summaries** validates our "knowledge pre-digestion" strategy, showing that a high signal-to-noise ratio in the context is crucial for unleashing the LLM's full reasoning potential.

## 4.4 CASE STUDY (RQ3)

A case study (Table 4) reveals the flaws of existing paradigms: **Naive RAG**'s one-shot" retrieval prevents multi-hop reasoning, while **GraphRAG** suffers from entity saturation," creating a noisy context that buries the answer and burdens the LLM.

In contrast, **BridgeRAG** demonstrates an efficient and precise path. It leverages planning and query refinement to define its objective, then uses a SAME_AS link as a surgical bridge between knowledge silos. The resulting context is clean and focused, proving that its dynamic, lightweight navigation overcomes the limitations of both **Naive RAG**'s incompleteness and **GraphRAG**'s contextual noise.

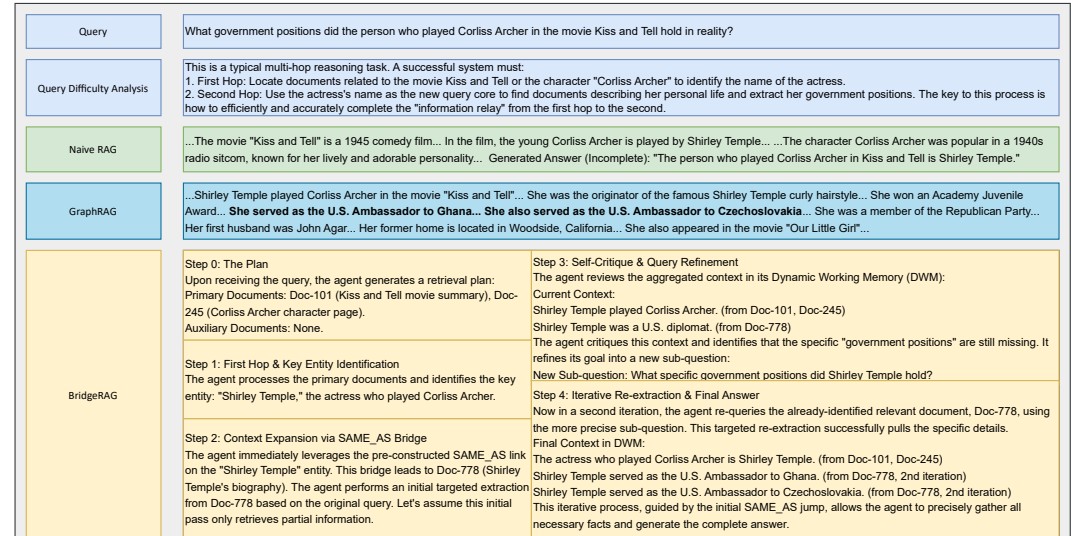

Figure 4: **Case Study of Reasoning Paths for Different RAG Paradigms.** BridgeRAG success-fully navigates the multi-hop query by leveraging a pre-constructed SAME_AS link to bridge knowl-edge partitions, avoiding the incompleteness of Naive RAG and the contextual noise of GraphRAG.

## 4.5 HYPERPARAMETER EXPERIMENTS (RQ4)

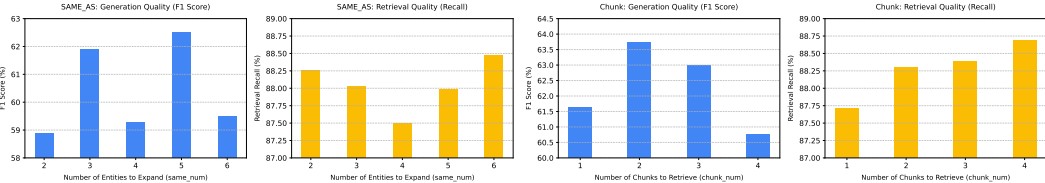

Figure 5: **Analysis of key online reasoning hyperparameters on the HotpotQA-Subset.** Anal-ysis of online reasoning hyperparameters. **(a, b)** The impact of the SAME_AS expansion degree (`expansion_degree`), which controls how many linked entities are used for context expansion. **(c, d)** The impact of the chunk retrieval number (`chunk_num`), which determines how many text chunks are extracted from documents during information retrieval.

**Figure 5** provides a compelling visual summary of our online reasoning experiments. Subplots **(a, b)** clearly illustrate the trade-off for `expansion_degree`, showing the F1 score peaking at 5, thus avoiding both an overly "barren" context (degree <5) and a "bloated" one (degree >5). Subplots **(c, d)** validate the "less is more" principle for `chunk_num`, with the F1 score maximized at 2 (63.74%) before dropping as additional chunks introduce noise and dilute focus. The figure provides decisive visual support for our final configuration.

## 5 CONCLUSION

This paper introduces BridgeRAG, a novel "planned navigation" framework that resolves the funda-mental trade-off between contextual purity ("partitioned isolation") and noise ("monolithic fusion") in multi-document KG-RAG. By converting static graph traversal into a dynamic "plan-extract-iterate" process, our framework achieves state-of-the-art results on benchmarks like HotpotQA. While acknowledging limitations such as reliance on the base LLM and a computationally intensive offline phase, we propose future directions, including integrating Hypergraphs to model higher-order relationships Kim et al. (2024); Lu et al. (2023) and employing specialized Graph LLMs or Agents to enhance navigational precision and efficiency Tang et al. (2024); Zhang (2023).

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

## 6 APPENDIX

The performance of BridgeRAG is influenced by several key hyperparameters that govern both the offline knowledge construction and the online reasoning phases. To ensure the robustness and optimal performance of our framework, we conducted a series of comprehensive experiments to determine the optimal value for each critical hyperparameter. The following sections detail the methodology and results of this analysis, validating our final parameter choices on the HotpotQA-Subset.

### 6.1 IMPLEMENTATION DETAILS

All experiments were conducted on a server equipped with 4x NVIDIA RTX 3080 (80GB) GPUs. To ensure fairness and reproducibility, all models, including our BridgeRAG and the baselines, shared a unified configuration for their core components. The specific fixed settings are as follows:

- **Base Large Language Model:** For all generative and reasoning tasks, including knowledge summarization, entity linking adjudication, retrieval planning, and final answer synthesis, we utilized `Qwen2.5-14B-Instruct`.
- **Text Embedding Model:** For all semantic similarity calculations, such as initial entity screening and online routing, we employed `nomic-ai/nomic-embed-text-v1.5`.
- **Document Chunking:** During the offline knowledge base construction, documents were segmented into chunks of 512 tokens with a 64-token sliding window overlap.
- **Max Iteration:** The maximum number of reasoning iterations in the online inference loop was set to 3.

### 6.2 HYPERPARAMETER ANALYSIS

To determine the optimal values for key hyperparameters in our BridgeRAG framework, we conducted a series of ablation studies on the HotpotQA-Subset. The following sections detail the experimental process and results for each parameter.

#### 6.2.1 KNOWLEDGE BASE CONSTRUCTION PHASE ($S_{LLM}$ AND LINK_THRESHOLD)

In our hybrid entity linking mechanism, we balance the deep semantic adjudication from the LLM ($S_{LLM}$) with the vector similarity from the embedding model ($S_{emb}$). To find the optimal balance, we analyzed the weight $\alpha$ assigned to $S_{LLM}$, where the final score is calculated as $\alpha \cdot S_{LLM} + (1 - \alpha) \cdot S_{emb}$. We varied $\alpha$ from 0.4 to 1.0.

Table 3 presents the end-to-end performance for different values of $\alpha$. Peak performance, as indicated by the primary metrics of F1 and SUB_EM, is achieved at $\alpha = 0.7$. The results reveal a clear trade-off: a lower $\alpha$ ($\leq 0.6$) over-relies on embedding similarity, which, despite its robustness, lacks the nuanced understanding required for precise linking. Conversely, a higher $\alpha$ ($\geq 0.8$) marginalizes the embedding signal, making the process susceptible to potential LLM hallucinations or misjudgments. The configuration of $\alpha = 0.7$ strikes an effective balance, leveraging the LLM's superior

reasoning while using the embedding score as a strong regularizer. We therefore adopted this value in our final configuration.

Table 3: **Hyperparameter analysis for the LLM score weight ($\alpha$) in the hybrid entity linking mechanism.** Performance is evaluated on the HotpotQA-Subset. The **bold** values denote the best results for each metric.

| $S_{LLM}$ ($\alpha$) | ROUGE-L | SUB_EM | TOKEN_PRECISION | TOKEN_RECALL | F1 | Retrieval Precision | Retrieval Recall |
|---|---|---|---|---|---|---|---|
| 0.4 | 50.12% | 55.65% | 63.24% | 64.77% | 61.98% | **75.83%** | 88.70% |
| 0.5 | 48.86% | 56.52% | 63.69% | 65.47% | 62.10% | 75.30% | **88.77%** |
| 0.6 | 51.03% | 56.52% | 63.72% | 64.83% | 62.42% | 75.48% | 87.79% |
| 0.7 | 52.24% | **59.13%** | **65.70%** | **68.48%** | **64.68%** | 75.13% | 88.17% |
| 0.8 | **52.32%** | 56.52% | 64.82% | 66.48% | 63.40% | 75.30% | 88.33% |
| 0.9 | 47.04% | 53.91% | 59.24% | 61.66% | 58.30% | 75.48% | 87.91% |
| 1.0 | 51.66% | **59.13%** | 63.96% | 67.79% | 63.09% | 75.13% | 88.17% |

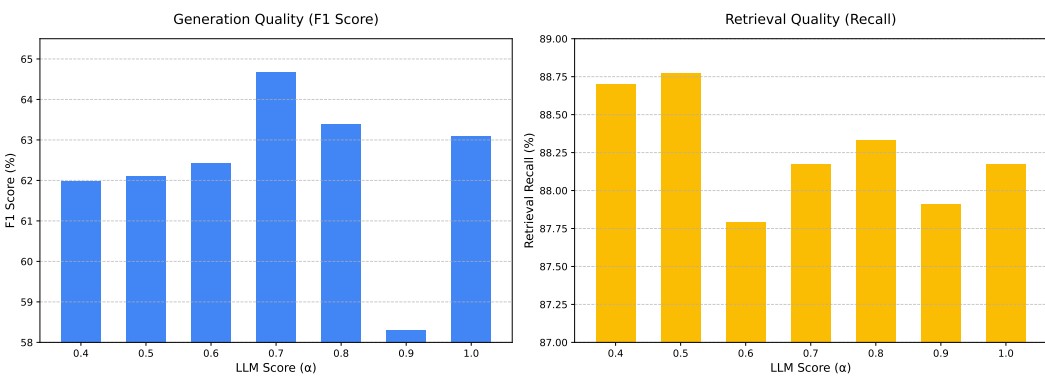

Figure 6: **Impact of the LLM adjudication score weight ($\alpha$) on generation quality (F1 Score) and retrieval quality (Recall) on the HotpotQA-Subset.** An optimal F1 score is achieved at $\alpha = 0.7$, highlighting the synergy between the LLM's deep adjudication and the embedding model's robust similarity signal.

The empirical results, detailed in Table 3, point to $\alpha = 0.7$ as the optimal weight for the LLM score. **Figure 6** provides a compelling visual representation of this finding. The bar chart for Generation Quality clearly shows the F1 score rising to a distinct peak at $\alpha = 0.7$ before declining. This visualization underscores the synergy of our hybrid approach: when the LLM's weight is too low ($\leq 0.6$), performance is capped by the embedding model's limitations. When it is too high ($\geq 0.8$), the model loses the robust grounding provided by the embedding signal. The figure demonstrates that the 0.7 weighting is not merely an incremental improvement but a significant sweet spot for model performance.

The linking threshold $\tau$ acts as the final gatekeeper for establishing a SAME_AS link, directly controlling the trade-off between the density (connectivity) and the precision (purity) of the cross-document knowledge graph. We systematically varied $\tau$ from 6.5 to 9.0 to identify its optimal value.

As shown in Table 4, the model's overall performance peaks at $\tau = 7.0$. This result highlights a critical balance. A lenient threshold (e.g., 6.5) increases connectivity but introduces noisy, incorrect links, leading to contextual pollution that degrades performance. In contrast, an overly strict threshold (e.g., $\geq 8.0$) prunes too aggressively, severing valid reasoning paths and resulting in an overly sparse knowledge graph. A threshold of $\tau = 7.0$ optimally balances link precision with network connectivity, ensuring sufficient pathways for multi-hop reasoning without introducing significant noise. This value was used for our main experiments.

The results in Table 4 reveal a clear performance peak at $\tau = 7.0$. This trend is visually corroborated in **Figure 7**, which plots the F1 score and Retrieval Recall against the linking threshold. The figure

Table 4: **Hyperparameter analysis for the linking threshold ($\tau$) in the hybrid entity linking mechanism.** Performance is evaluated on the HotpotQA-Subset. The **bold** values denote the best results for each metric.

| link_threshold ($\tau$) | ROUGE-L | SUB_EM | TOKEN_PRECISION | TOKEN_RECALL | F1 | Retrieval Precision | Retrieval Recall |
|---|---|---|---|---|---|---|---|
| 6.5 | **53.05%** | 58.41% | 65.07% | 65.98% | 63.91% | **75.93%** | 88.94% |
| 7.0 | 52.24% | **59.13%** | **65.89%** | **68.48%** | **64.68%** | 75.13% | **89.02%** |
| 7.5 | 49.93% | 56.52% | 63.02% | 65.76% | 61.96% | 75.30% | 88.58% |
| 8.0 | 51.11% | 56.52% | 63.97% | 64.30% | 62.52% | 75.30% | 87.47% |
| 8.5 | 50.77% | 56.52% | 64.41% | 65.66% | 62.87% | 74.96% | 88.56% |
| 9.0 | 48.70% | 53.91% | 62.19% | 62.76% | 60.88% | 75.83% | 89.01% |

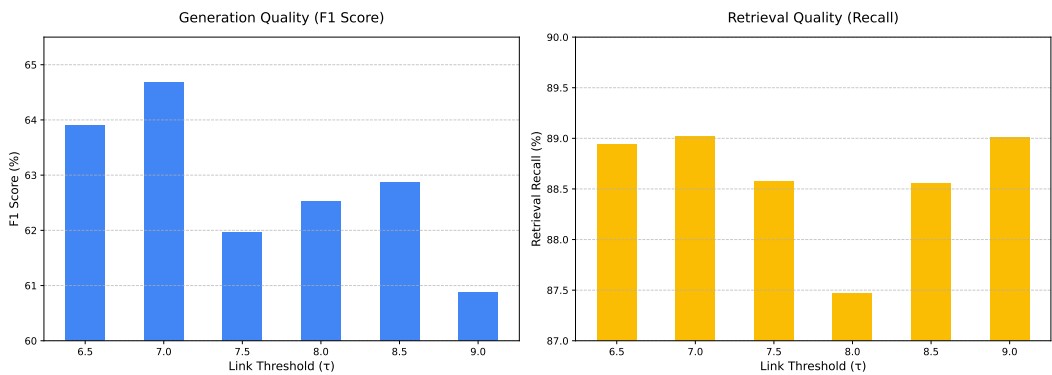

Figure 7: **Impact of the linking threshold ($\tau$) on generation quality (F1 Score) and retrieval quality (Recall) on the HotpotQA-Subset.** The results show that a threshold of $\tau = 7.0$ achieves the highest F1 score, striking an optimal balance between link precision and network connectivity.

illustrates a critical trade-off: a lenient threshold (e.g., 6.5) maintains high recall but at the cost of introducing erroneous links that pollute the context, thereby failing to achieve the highest F1 score. Conversely, an overly strict threshold (e.g., $\geq 8.0$) causes a sharp drop in both metrics by aggressively pruning valid reasoning paths. The visualization clearly confirms that $\tau = 7.0$ represents the optimal balance, ensuring sufficient network connectivity for reasoning while maintaining high link precision.

### 6.2.2 ONLINE ITERATIVE REASONING PHASE (`EXPANSION_SCOPE` AND `CHUNK_NUM`)

During online inference, the `expansion_scope` parameter determines how many linked entities are expanded to enrich the context. This directly controls the breadth of information available for reasoning. We tested values from 2 to 6 to find the best configuration.

The results in Table 5 indicate that the optimal performance is achieved when `expansion_scope` is set to 5. This demonstrates a clear trade-off between information gain and contextual noise. Insufficient expansion (`expansion_scope` <5) results in a context that is too "barren" to fully support complex reasoning chains. Conversely, excessive expansion (`expansion_scope` >5) inundates the context with potentially irrelevant information, creating a "bloated" context that can distract the LLM. Expanding the top 5 most relevant entities provides a sufficiently rich yet focused context for the agent.

The `chunk_num` parameter dictates how many relevant text chunks are retrieved from each document to form the final context. This hyperparameter governs the granularity and density of the evidence presented to the LLM. We evaluated values from 1 to 4.

Table 6 shows that retrieving the top 2 chunks yields the best overall performance (F1: 63.74%). This finding supports a "less is more" principle for context construction. Retrieving only one chunk often provides insufficient evidence, while retrieving three or more introduces noise and dilutes the

Table 5: **Hyperparameter analysis for the number of SAME_AS entities to expand (`expansion_scope`) during the online inference phase.** Performance is evaluated on the HotpotQA-Subset. The **bold** values denote the best results for each metric.

| expansion_scope | ROUGE-L | SUB_EM | TOKEN_PRECISION | TOKEN_RECALL | F1 | Retrieval Precision | Retrieval Recall |
|---|---|---|---|---|---|---|---|
| 2 | 48.33% | 53.91% | 59.62% | 61.81% | 58.90% | 75.30% | 88.26% |
| 3 | 49.07% | 55.65% | 62.93% | **64.35%** | 61.90% | **75.65%** | 88.03% |
| 4 | 47.55% | 54.78% | 59.92% | 62.34% | 59.28% | 75.49% | 87.50% |
| 5 | **51.11%** | **56.52%** | **63.97%** | 64.30% | **62.52%** | 75.30% | 87.99% |
| 6 | 48.74% | 53.91% | 60.21% | 61.90% | 59.49% | 74.96% | **88.47%** |

focus on the most critical information, potentially distracting the LLM. Therefore, `chunk_num=2` was chosen as it strikes the ideal balance between informational completeness and contextual purity.

Table 6: **Hyperparameter analysis for the number of chunks to retrieve (`chunk_num`) during the online information extraction phase.** Performance is evaluated on the HotpotQA-Subset. The **bold** values denote the best results for each metric.

| chunk_num | ROUGE-L | SUB_EM | TOKEN_PRECISION | TOKEN_RECALL | F1 | Retrieval Precision | Retrieval Recall |
|---|---|---|---|---|---|---|---|
| 1 | 48.61% | 55.65% | 62.70% | 64.59% | 61.64% | 75.65% | 87.72% |
| 2 | **52.00%** | **57.39%** | **64.94%** | **65.80%** | **63.74%** | **75.65%** | 88.30% |
| 3 | 49.89% | **57.39%** | 63.74% | 65.18% | 62.98% | **76.00%** | 88.39% |
| 4 | 50.35% | 54.78% | 61.75% | 63.38% | 60.77% | 75.38% | **88.69%** |

