# OpenReview forum: "BridgeRAG: A Framework for Reasoning over Partitioned Knowledge Graphs"
_ICLR.cc/2026/Conference — Submitted to ICLR 2026_

### Official Review · Reviewer_biM7 · 2025-10-28

**Soundness:** 1
**Presentation:** 1
**Contribution:** 2
**Rating:** 0
**Confidence:** 4

**Summary:**

The paper BridgeRAG: A Framework for Reasoning over Partitioned Knowledge Graphs presents a framework for querying over multiple documents, evaluated in the context of multi-hop question answering. The documents are first transformed into knowledge graphs using an NLP pipeline to extract named entities as well as relations. However, the system does not use graphs per se, as the LLMs are not equipped for dealing with data in graph format. Therefore, the resulting graph is summarized using an LLM (both at the entity level and at the document level), and the rest of the processing is done on the generated text.

The authors of the paper do not clearly delineate their own work from existing tools and methods. It is therefore very difficult to accurately delineate their contributions.

The contributions as stated by the authors consist in proposing a framework for doing RAG across multiple documents and introducing three mechanisms to this end: knowledge predigestion (creating entity-level and document-level summaries based on the information in the knowledge graph); hybrid entity linking (matching the entities across documents using encodings of their summaries and an LLM-based SAME_AS score) and dynamic working memory (which deals with expanding the graph starting from the original query and documents).

**Strengths:**

The paper has potential, however, the way it was written makes it difficult to understand and to replicate. For example, the idea of creating a document manifest, i.e. a list of all entities followed by a document summary is interesting. The planning also seems to matter according to the ablation study, but unfortunately it is not described in a way that can be understood.

**Weaknesses:**

The paper is written in a tabloid style, with unnecessarily bombastic wording, making it more difficult to understand what the authors describe. The authors should revise their prose and stick to the scientific style of writing, focusing on describing facts and offering clear explanations.

Some terms are used throughout the paper without ever being properly introduced: e.g. entity saturation, contextual noise pollution, contextual purity. Adding quotes around a term does not explain it.

The paper is difficult to read, as authors do not use citations properly: please use \parencite{smith2020} -> (Smith, 2020) when the citations are not part of a sentence. And \textcite{smith2020} -> “Smith (2020)…” when the citation is part of a sentence.

Moreover, the authors do not clearly delineate their own contribution from the different off-the-shelf tools they use.

The methodology section is written extremely superficially, without mentioning important aspects about the framework being proposed. The explanations offered are mostly handwavy, and are of no use for understanding the methods.

I would recommend a careful rewrite of the paper, based on the questions below.

**Questions:**

In Fig. 1, subfigure 3 the bridged graph can easily become a monolithic graph – it only needs a couple of links added for all nodes to be completely accessible from any node of the graph. Which begs the question: how do you test/ensure that the connectivity in the graph remains bridge-style and does not become monolithic?

The methodology is unclear:
-	What information about an entity is used to generate the entity summary?
-	How is N_1(e) defined?
-	What prompt is used to generate the summary?
-	Which LLM was used in the experiments? (this appears only in the appendix)
-	What prompt is used to generate the document summary? Is the document summary more than a concatenation of the individual entity summaries?
-	How are the Partitioned Knowledge Graphs created? Which knowledge based was used?
-	What is a hybrid entity linking mechanism? What entity linker was used? Is this an off-the-shelf entity linker?
-	What is a multi-source weighted router? Is this an existing tool or something that you developed?
-	What termination condition is used for the guided retrieval?
-	Which independent NLP pipeline is used in S. 3.2.1? Is this something the authors created, or an existing library? If it is an existing library, please add proper citations.
-	In Section 3.2.1, the KG construction is unclear: is the knowledge graph G_i constructed only based on the information from the document, d_i, or is there other external knowledge from a knowledge base that is used (maybe which is part of the NLP pipeline)?
-	In section 3.2.2, I am assuming that the advanced sentence embedding model is, again, a third-party library which is not properly cited.
-	In the deep semantic adjudication section, how is the output of the LLM calibrated – e.g. how is the scoring mechanism trained?
-	Why not use directly an entity linker (e.g. BLINK, https://github.com/facebookresearch/BLINK) rather than this ad-hoc hybrid approach?
-	In Section 3.3.1, what is the Reciprocal Rank Fusion algorithm? Is this something the authors designed, or existing work?
-	In Section 3.3.1, what is the algorithm/method for deciding if a document is primary or auxiliary?
-	In Section 3.3.2, what is the difference between the mining strategy for primary and for auxiliary documents? It is not explained at all.
-	In Section 3.3.3., what is a critique function?
-	In Section 3.3.3, how does the agent formulate a precise sub-question?
-	In Section 4.1, where does the 2WikiMultiHopQA-Subset come from?
-	In Section 4.1, the evaluation metrics are atypical for a question answering task. The authors must present it in more detail.

---

### Official Review · Reviewer_bDUS · 2025-11-03

**Soundness:** 3
**Presentation:** 2
**Contribution:** 2
**Rating:** 4
**Confidence:** 3

**Summary:**

The paper tackles a key pain point in KG-based RAG for real-world, multi-document settings: if you build a separate KG for each document, you keep context clean but can’t answer questions that span documents; if you merge everything into one big KG, you get entity collisions (“Apple” vs “apple”) and noisy retrieval. To resolve this “contextual purity vs knowledge connectivity” dilemma, the authors propose BridgeRAG, a two-phase framework. Offline, it “pre-digests” documents and uses hybrid (LLM + embedding) entity linking to build trustworthy SAME-AS links across otherwise isolated document-level KGs. Online, an LLM agent does iterative, multi-hop reasoning over these linked partitions: a multi-source weighted routing module picks the most relevant documents, and a Dynamic Working Memory (DWM) keeps only the highly relevant facts in context. Experiments on multi-hop QA show BridgeRAG retrieves cleaner cross-document evidence and beats prior KG-RAG setups.

**Strengths:**

1.the paper clearly formulates the core KG-RAG problem as “partitioned isolation vs cross-partition linking,” which is exactly what happens in multi-document enterprise / report / case-file settings. It combines offline high-fidelity entity linking (to avoid KG pollution) with online, agentic, step-by-step reasoning + DWM is a nice division of labor and more realistic than doing everything at query time. The shared named entities are the main “bridges” between documents is intuitive, inspectable, and lets them control noise better than naïve graph merging.

2.Empirical results have shown the effectiveness of the proposed approach.

**Weaknesses:**

1.if NER/coref/linking is wrong or sparse (domain jargon, long-tail entities), the “bridges” won’t form and cross-document reasoning degrades. It means that the proposed approach is heavily relying on the correction of intermediate steps. It should be analyzed that whether error accumulation or potential risk in real-world scenarios.

2.the offline “knowledge pre-digestion” + dual-verification linking step is extra machinery that must be re-run when documents change, which may limit use in highly dynamic corpora. It is also time-consuming in practice. More latency analysis should be done to show the efficiency problem.

3.Entity-as-bridge is a strong assumption. Some cross-document reasoning relies on events, schemas, or implicit relations not surfaced as named entities. The current design might under-retrieve those cases unless extended with relation/schema-level linking.

**Questions:**

Please refer to the weakness part

**Details Of Ethics Concerns:**

Please refer to the weakness part

---

### Official Review · Reviewer_MExF · 2025-11-09

**Soundness:** 2
**Presentation:** 2
**Contribution:** 2
**Rating:** 4
**Confidence:** 4

**Summary:**

The paper “BridgeRAG: A Framework for Reasoning over Partitioned Knowledge Graphs” presents a RAG architecture for multi-document reasoning. It builds isolated knowledge graphs for each document and links them through a hybrid entity-linking mechanism combining embeddings and LLM verification. During inference, an agent performs iterative retrieval and reasoning using a Dynamic Working Memory (DWM) that refines queries into sub-questions when information is insufficient.

**Strengths:**

Strength: (1)Comprehensive Framework Integration: BridgeRAG effectively combines multiple components—entity linking, document summarization, and iterative reasoning—into a cohesive architecture that addresses both contextual isolation and cross-document connectivity in multi-document RAG.

(2)Strong Empirical Performance: The framework demonstrates notable improvements on multi-hop question answering benchmarks, suggesting its potential for enhancing reasoning accuracy and retrieval precision in complex knowledge integration tasks.

**Weaknesses:**

Weakness: (1) In the Partitioned KG Construction and Pre-Digestion stage, each document contains a large number of entities, and the total number across documents is even higher. Invoking the LLM to summarize every entity neglects the issue of computational cost, leading to significant resource waste as the number of KG nodes increases. It is recommended to include experiments analyzing time consumption and computational overhead. Moreover, while the idea of leveraging broad world knowledge and contextual cues from entity summaries to disambiguate whether two entities refer to the same real-world object is sound, this approach causes severe exponential growth in LLM invocations, making it impractical for large document collections.

(2) The Dynamic Working Memory (DWM) module in BridgeRAG is presented as the “core innovation” of the framework, but from a rigorous review perspective, its novelty is limited. The first step of DWM—using IsSufficient to determine whether the current context adequately answers the question—cites Liu et al. (2025) and follows the established “self-evaluation and reflection” paradigm seen in Self-RAG (Asai et al., 2024), Reflexion (Shinn et al., 2023), and Critic-CoT / Self-Consistency (Kumar et al., 2025). The subsequent “generation of new sub-questions” is likewise a common Query Refinement strategy. The second step,
, which generates refined sub-questions based on prior context, is already well-established, as in RQ-RAG (Chan et al., 2024) under its “query refinement through LLM planning” approach.

(3) The construction of cross-document experimental datasets is insufficiently described. Given that typical LLMs handle around 5,000 tokens effectively, the rationale for the chosen dataset configuration remains unclear. It would be more convincing to include cross-document evaluations involving longer contexts—e.g., 100,000-token settings—to justify the dataset design.

**Questions:**

(1)Regarding computational efficiency: How does BridgeRAG handle the significant computational overhead introduced by invoking the LLM to summarize every entity and perform dual-verification linking across documents? Could the authors provide quantitative analyses of time and resource consumption to demonstrate scalability on large knowledge graphs?
(2)On methodological originality: Given that the Dynamic Working Memory (DWM) module closely follows established self-reflection and query refinement paradigms (e.g., Self-RAG, RQ-RAG), what concrete methodological innovations distinguish BridgeRAG’s DWM from these prior approaches beyond architectural integration?
(3)About dataset construction and scalability: Could the authors elaborate on the design rationale for the chosen multi-document datasets and provide evidence that BridgeRAG remains effective in much larger or longer-context settings (e.g., >100K tokens), where cross-document reasoning becomes more challenging and realistic?

---

### Meta-Review · Area_Chair_Apud · 2025-12-20

**Summary:**

The paper addresses a core challenge of KG-based RAG in realistic multi-document settings. Building a separate knowledge graph for each document preserves contextual clarity but prevents answering cross-document questions, while merging everything into a single global KG introduces entity collisions  and noisy retrieval. To overcome this trade-off between contextual purity and knowledge connectivity, the authors propose BridgeRAG, a two-phase framework. In the offline phase, documents are pre-processed and hybrid entity linking  based on LLMs and embeddings is used to establish reliable SAME-AS links across otherwise isolated document-level KGs. In the online phase, an LLM agent performs iterative multi-hop reasoning over these linked partitions: a multi-source weighted routing module selects the most relevant documents, and a Dynamic Working Memory retains only highly relevant facts in context. Experiments on multi-hop question answering show that BridgeRAG retrieves cleaner cross-document evidence and outperforms prior KG-RAG approaches.

**Reviewer Concerns:**

While all reviewers see potential in the idea there are severl points in which the current manuscript can be improved. Main criticisms concern scalability,  robustness, clarity, and real-world applicability.  More precisely, scalabilty and efficienty concerns are that the offline pre-digestion phase (entity summarization and cross-document linking) requires a large number of LLM calls. Runtime, latency, and resource usage are not analyzed, raising doubts about practicality for large or dynamic corpora. Morever, the the proposed approach could have several issues in practice: 1. It heavily relies on accurate NER, coreference resolution, and entity linking. Errors or sparsity in these steps (e.g., domain-specific or long-tail entities) can prevent cross-document “bridges” from forming and lead to error accumulation. 2. Treating named entities as the primary cross-document bridges may miss reasoning that depends on events, schemas, or implicit relations, limiting generality and recall. It is not clear how this points effect the performance of the proposed approach.
Also the expermental part and the writing could be improved: Key implementation details (KG construction, prompts, models, routing, termination conditions, scoring mechanisms) are missing or vaguely described. Dataset construction and evaluation choices are not well explained, there is no validation in larger or longer-context settings, and a limited analysis of the graph connectivity.

**Reviewer Scores:**

The authors did not get involved in the rebuttal, so it is not clear, if they see that the listed points can be improved. Therefore, I do not expect that the reviewers would have changed their scores.

---

### Decision · Program_Chairs · 2026-01-26

Reject